# Construction Disputes in the UAE: Causes and Resolution Methods

**Sameh El-Sayegh**[ID]**, Irtishad Ahmad \*, Malak Aljanabi, Rawan Herzallah, Samuel Metry and Omar El-Ashwal**

Civil Engineering Department, American University of Sharjah, Sharjah P.O. Box 26666, UAE; selsayegh@aus.edu (S.E.-S.); g00061357@aus.edu (M.A.); g00058017@aus.edu (R.H.); b00056992@aus.edu (S.M.); B00054449@aus.edu (O.E.-A.)
\* Correspondence: irahmad@aus.edu

**Abstract:** Claims and disputes occur frequently in the construction industry between different contracting parties, mainly the owner, the designer and the contractor. Consequently, valuable time and a significant amount of money are lost. The United Arab Emirates (UAE) construction industry, one of the most vibrant sectors globally, is experiencing a high level of construction disputes and claims. This paper aims to identify and assess the major causes of disputes in the UAE and weigh the effectiveness of the methods used for their avoidance and resolution. The sources of disputes, and their avoidance/resolution methods, were identified through a comprehensive literature review. A survey was then developed and sent to 150 construction professionals. Fifty-four responses were received and analyzed. The results show that the top five sources of disputes in the UAE are variations initiated by the owner, obtaining permit/approval from the municipality and other governmental authorities, material change and approval during the construction phase, the slowness of the owner in decision-making, and the short time available during the design phase. As for the avoidance and the resolution method, the most effective method was found to be negotiation.

**Keywords:** disputes; claims; dispute resolution methods; construction industry; UAE

## 1. Introduction

Disputes are common in the construction industry. Disputes arise due to disagreements between any of the contracting parties. Disputes have a devastating effect on construction projects, as they may result in cost overruns, delays, and loss of productivity. It is vital to understand the causes of disputes to complete a construction project within cost and time [1]. Construction disputes impact project objectives and strain relationships between contracting parties [2]. A dispute in construction projects is considered to be an impediment to the path of successful project completion [3]. Disputes are resource consuming, unpleasant, and expensive [4]. Conflicts disrupt the flow of work, resulting in additional costs, delays, and other negative impacts [5]. These problems might lead to construction claims and disputes. Arditi and Pattanakitchamroon [6] discussed various reasons for such problems, including incorrect design/specifications, unusually severe weather condition, change orders and extra work. Disputes mainly arise from claims resulting from differing site conditions, delays, design errors or changes, acceleration or suspension of work, construction failures, and additional or deleted work [7]. Mitropoulos and Howell [8] suggested that the main factors that influence disputes are project uncertainty, opportunistic behavior, and contractual problems.

Disputes, if not addressed and resolved properly, can create significant losses in the project and for the company [9]. Artan and Bakioglu [10] argued that risk is a major factor leading to disputes. Cheung and Pang [11] divided construction disputes into two types based on the sources: contractual

and speculative. The main causes of both types of construction disputes are ambiguity, deficiency, inconsistency, and defectiveness, which are grouped under the "contract incompleteness" [11]. They further identified factors such as risk, uncertainty and conflicts as other causes of contractual disputes. On the other hand, human perception factors such as ambiguity of contractual clauses that may cause interpretational difficulties, opportunistic behavior and conflicts could be other causes of disputes [11].

The problem of disputes and claims plagues the construction industry. The leading causes need to be identified in specific regions, since each region has a unique setting due to their differing legal, political and cultural aspects. The identification of causes will lead to appropriate resolution methods being chosen depending on the specific case. In this paper, the country investigated is the United Arab Emirates, which has a very vibrant economy due to growing tourism and business activities. As a consequence, there is a continuous growth in the construction industry in the United Arab Emirates (UAE), which attracts international contractors and investors to its large and unique projects. Construction disputes have increased in recent years due to the increase in the size and complexity of construction projects. Awwad et al. [5] found that, in the Middle East, the UAE is the second highest country in terms of the value of construction disputes after the Kingdom of Saudi Arabia (KSA). Moreover, the Middle East has the longest average duration of dispute settlement, which is 14.6 months [5]. The UAE construction industry is experiencing fast growth with many ongoing projects, investment into green open spaces and strong government support. The construction industry contributes more than 10% to the United Arab Emirates' (UAE) gross domestic production (GDP). The UAE construction industry is growing consistently and expected to grow more in the near future to accommodate the UAE's strategic goals, which include significant spending in infrastructure construction. The construction activity is projected to continue rising as a percentage of real GDP in the UAE; from 10.3% in 2011 to 11.5% in 2021 [12]. The growth of the construction industry in the UAE is accompanied by an increase in claims and disputes, which, as a result, leads to delays, and additional cost to the project.

The two most common ways to mitigate disputes are avoidance and resolution. Resolution methods are further divided into early and late categories. Negotiation, risk allocation, early non-binding neutral evaluation, and partnering are included under avoidance methods. Early resolution methods included negotiation, conciliation, and mini-trial/executive tribunal. Late resolution methods included, negotiation, arbitration, mediation, adjudication, dispute review boards, and litigation.

Studies show that the most frequently used methods to resolve disputes are negotiation, arbitration, and litigation. Alternative Dispute Resolution (ADR) methods proved to be more efficient in settling disputes, with less cost and time compared to those most used in the Middle East [5]. The Hong Kong International Arbitration Center (HKIAC) stated that increasingly complicated construction projects over the last two decades have resulted in a growing number of projects adopting Alternative Dispute Resolution (ADR) [9]. The construction industry has called for alternatives owing to the high cost, in terms of time and money, of taking things to a court or organizing for arbitration [9]. Martin and Thompson [13] discussed the various methods of dispute resolution and provided tools that parties can use to manage the selected process of dispute resolution. Martin and Thompson [13] identified five basic forms of construction dispute resolutions, which are collaborating, Dispute Review Board (DRB), mediation, arbitration, and litigation. According to Yates [7], there are different traditional resolution techniques such as arbitration, litigation, and negotiation. The problem with litigation and arbitration is that both of them are adversarial processes, in other words, one side wins and the other side lose [7]. Using adversarial methods could weaken relationships between the two sides involved [7]. Cheung et al. [14] found that negotiation between the disputants takes first place in resolving any problem. Cheung et al. [14] stated that the styles of negotiation depend on the personality of the disputants, situation volume and type. Negotiation helps to prevent members of different construction firms from using undesirable methods such as arbitration or litigation; in addition, it saves time, money and the reputation of the firm [7]. Mitropoulos and Howell [4] showed that the prevention of high-cost,

complicated disputes mainly depends on the problem-solving ability and planning of the project organization. The main methods used to resolve the claims in the UAE are negotiation, meditation, arbitration, and litigation [15]. Ho and Liu [16] stated that claims and disputes are integrated and interrelated. Ho and Liu [16] developed a Model based on the "Game Theory" or "Nash-Equilibrium" that helps the owner to avoid the presence of construction claims due to opportunistic bidding. Haugen and Singh [9] identified three relevant impacting factors in deciding the ADR method: The market position of the individual parties, the relationship between parties, and the source of dispute.

The main objectives of the paper are twofold. First, to identify and present the major causes and sources of disputes and claims in the UAE construction industry. Second, to outline the effective methods of avoidance and resolution. The major causes and leading methods of mitigation are identified from the available literature and publications. These findings provided the basis of a survey instrument conducted among the construction industry professionals in UAE. The method used is described in the next section.

## 2. Materials and Methods

In order to accomplish the objectives of this research study, the first step undertaken was a thorough review of the existing literature on the subject. This helped researchers to identify the main sources of construction disputes and their avoidance and resolution methods. Twenty-seven causes of disputes were identified. They are grouped in five categories: Design-related, owner-related, contractor-related, contractual, and 'other' disputes. The literature review was the basis of a survey instrument that was developed to be distributed to the UAE construction professionals.

The survey included three sections. The first section obtained basic information about the profile of the respondents. This included questions about the company type, the company's years of experience, industry sector (type), the company type, and the size of the projects undertaken by the company. The second section identified the major causes of disputes based on their frequency of occurrence. Respondents were asked to use the Likert Scale with the ratings: very high, high, moderate, low, and very low. The third section measured the effectiveness of dispute resolution and avoidance methods as practiced in the UAE by using the ratings extremely effective, effective, neutral, not effective, extremely not effective and not applicable. The survey was sent to 150 construction professionals. Fifty-four responses were received, corresponding to a 36% response rate. Out of the 54 responses, 28 were local companies and 26 were international companies. Table 1 shows the respondents' profile.

**Table 1.** Respondents' Profile.

| Category | | Respondents (54) | |
|---|---|---|---|
| | | Number | % |
| Years of experience | >20 years | 9 | 16.7% |
| | 11–20 years | 16 | 29.6% |
| | 5–10 years | 20 | 37.0% |
| | <5 years | 9 | 16.7% |
| Role | Owner | 4 | 7.4% |
| | Consultant | 16 | 29.6% |
| | Contractor | 29 | 53.7% |
| | Management Consultant | 5 | 09.3% |
| Average Project Size [1] | <50 (Million AED) | 8 | 14.8% |
| | 50–200 (Million AED) | 15 | 27.8% |
| | 200–500 (Million AED) | 8 | 14.8% |
| | >500 (Million AED) | 23 | 42.6% |

[1] 1 USD = 3.67 AED (2020 Currency).

The survey results are analyzed to identify the frequently occurring disputes in the UAE. The weighted average is calculated for each of the disputes and resolution/avoidance method. The weight is assigned for each dispute source, based on frequency of occurrence, as 5, 4, 3, 2, and 1 for very high, high, moderate, low and very low, respectively. Likewise, the weight assigned for the effectiveness of each resolution/avoidance method is 5, 4, 3, 2, 1, and 0 for extremely effective, effective, neutral, not effective, extremely not effective and not applicable, respectively. The Weighted Average (WA) is calculated using Equation (1)

$$\text{Weighted Average, } WA = \frac{\sum_{i=1}^{5} W_i \times X_i}{\sum_{i=1}^{5} X_i} \tag{1}$$

where:

$W_i$ = Weight assigned to $i$th response; $W_i$ = 1, 2, 3, 4 and 5 for $i$ = 1, 2, 3, 4 and 5, respectively;

$X_i$ = Frequency of the $i$th response;

$i$ = Response category index = 1, 2, 3, 4 and 5 for very low, low, moderate, high and very high, respectively.

Equation (1) is also used for the effectiveness of the dispute resolution methods, but the response category index ($i$) was assigned 5, 4, 3, 2, 1, and 0 for extremely effective, effective, neutral, not effective, extremely not effective and not applicable, respectively. The causes of disputes are ranked based on their weighted average scores. The cause with the highest average is ranked 1, and so on. The same process is used to rank the dispute resolution methods; however, the ranking is within each group only. For comparison purposes and to study the strength of relationship between two sets of ranking, the Spearman rank correlation coefficient (RHO) was determined using the IBM SPSS 26 software (IBM corporation, Armonk, NY, USA). A higher value of RHO (approaching 1) indicates a strong correlation.

## 3. Sources of Construction Disputes

Twenty-seven sources of disputes were identified through the literature review and divided into five groups: designer, owner, contractor, contractual and other. Table 2 lists the sources of disputes along with the references.

**Table 2.** Sources of Disputes suggested by the literature.

| | Sources of disputes | Literature Source |
|---|---|---|
| **Designer-related** | Time limitation in the design phase | [17–22] |
| | Poor design | [23] |
| | Inadequate or incomplete technical plans/specification | [5] |
| | Poor preparation and approval of drawings | [15,17,24–32] |
| | Material change and approval during the construction phase | [15,17–19,22,24,33,34] |
| **Owner-related** | Slowness of the owner's decision-making process | [17,24] |
| | Inadequate early planning of the project | [17,24] |
| | Failure to make interim awards on extensions of time and compensating by the owner | [5] |
| | Variations initiated by the owner (additive/deductive) | [5] |
| | Poor Financing by the owner | [15,24,29,32] |
| **Contractor-related** | low Financing by the contractor during construction | [15,24,29,32] |
| | Shortage and unproductive workers | [24] |
| | Inadequate site investigation | [15,25–27,32] |
| | Poorly defined scope of work | [17,21,22,24,34–36] |
| | Poor supervision and site management | [17,24] |
| | Unsuitable leadership style of construction/project manager | [17–19,24,37–39] |
| | Underestimation and incompetence of contractors | [5] |
| **Contractual** | Poorly written contracts | [9,15,17–22,24,25,29,31,34,37,40] |
| | Differing Site Conditions | [23] |
| | Contract Amendments | [5,41] |
| | Contradictory and inaccurate information in the contract documents | [5] |
| **Other** | Obtaining Permit/Approval from the municipality/different government authority | [15,17–19,24,37,40,42,43] |
| | Modifying legislation and regulations | [5] |
| | Inappropriate weather conditions | [44] |
| | Impact on locality in terms of noise, traffic, and pollution/contamination | [45] |
| | Lack of communication and coordination between parties during construction | [17–19,21,24,33,34,40] |
| | impact of local cultures and social values in the settlement of conflicts | [5] |

### 3.1. Designer-Related Disputes

Time limitation in the design phase occurs as clients typically allow a very limited time to complete and submit the designs. If the design time is very limited, the design may lack specific details and accuracy. Poor design occurs when the design is nonfunctional, has missing elements and does not meet the owner's requirements. Inadequate or incomplete technical plans and specifications can cause delays during construction. Changes in material specifications and the subsequent approval process may take time, and cause disputes regarding the additional costs of the new materials and delays due to shipping and the difficulty procuring the material. All these cost the owner and designer, which, in turn, causes disputes [46].

### 3.2. Owner-Related Disputes

One of the most common causes of disputes is the slowness of the owner's decision-making process, in which the owner takes a long time making decisions, which delays the construction process. Inadequate early planning of the project creates disputes and causes additional cost to the owner [32]. Therefore, detailed early planning is required to avoid conflicts and extra costs. Failure to make interim awards on extensions of time and compensation by the owner is a common practice amongst owners. This may cause even more difficult-to-resolve disputes towards the end of the project. Waiting until the end of the project to deal with disputes makes them harder and costlier to resolve [47]. Owners may request variations such as adding or deducting from the previously agreed scope, which cause disputes since some variations may result in additional time and cost. If the owner cannot finance the project on time, construction will be delayed and the project might stop for a certain period until the owner is ready to finance the project, which creates disputes [8].

### 3.3. Contractor-Related Disputes

Inadequate financing by the contractor during construction leads to delays, work interruption, and poor quality of subcontractors' work. If the contractor has a shortage of workers, a delay in the building process occurs and this can create disputes. Disputes can be created due to low productivity [10]. Inadequate site investigation can create multiple disputes. A poorly defined scope of work is a type of contractor dispute. Poor supervision and site management by the contractor can cause accidents and delay the construction process. Site management is a major factor of construction disputes led by the contractor. Likewise, an unsuitable leadership style from the construction/project manager takes place when there is an incompetent person lacking appropriate qualifications in a position of construction/project manager, which results in them making the wrong decisions during construction [13]. Contractors can be asked to stop the construction process if the contracting company is not capable to continue to make progress in the project [48].

### 3.4. Contractual Disputes

A poorly written contract leads to different interpretations of the same issue, and that leads to an argument, which later on develops into a dispute. In addition, differing site condition is also considered a contractual dispute. This is when the contractor encounters unknown physical conditions of an unusual nature that differ materially from those that are ordinarily encountered and generally recognized as inherent in the work at the project's location [8]. Differing site conditions is the top reason for claims [6]. Contract amendments are used when the parties want to modify the terms of an existing legal agreement. Contradictory and inaccurate information in the contract document is when the contract needs to be very accurate and needs to be revised before an agreement between the parties is made. The statements in the contract should not contradict the scope of work and should be clarified properly between the parties.

### 3.5. Other Disputes

A source of other disputes is obtaining permits and/or approvals from the municipality/different government authority. Delays in getting approvals and permissions from official governmental offices

might lead to disputes. Modifying legislation and regulations can cause disputes. Each country has its own law and rules that are subject to change. These changes might cause disputes during the construction process. Similarly, inappropriate weather conditions can cause delays and cost overruns, which consequently result in disputes. Some construction projects might result in traffic jams due to road closure or noise, especially in areas that have schools or hospitals, for example, neighbors' complaints [5]. The lack of communication and coordination between parties during construction causes confusion and misunderstanding of the scope of work, and this causes social disputes between the parties.

## 4. Dispute Avoidance and Resolution Methods in Construction

Table 3 shows the dispute avoidance and resolution methods along with their literature sources.

**Table 3.** Summary of dispute avoidance and resolution methods.

| S/N | Method | Literature Source |
|---|---|---|
| Dispute Avoidance Methods | | |
| 1 | Negotiation | [5,9,23,45,49,50] |
| 2 | Risk Allocation | [5] |
| 3 | Early Non-Binding Neutral Evaluation | [5,51] |
| 4 | Partnering | [5,9] |
| Early Resolution Methods | | |
| 5 | Negotiation | [5,9,23,45,49,50] |
| 6 | Conciliation | [9,23,49,50,52] |
| 7 | Mini-Trial/Executive Tribunal | [23,52] |
| Late Resolution Methods | | |
| 8 | Negotiation | [5,9,23,45,49,50] |
| 9 | Arbitration | [23]; [49–51,53] |
| 10 | Mediation | [5,9,13,23,49–51,54] |
| 11 | Adjudication | [23,49,50,52] |
| 12 | Dispute Review Boards | [5,23,51,52] |
| 13 | Litigation | [5,9,13,51,52] |

### 4.1. Dispute Avoidance Methods

Dispute avoidance methods are used to prevent disputes from occurring. At the beginning of any claim, usually, all parties would choose the resolution methods that eliminate the dispute at its root [55]. Dispute avoidance methods include negotiation, risk allocation, early non-binding neutral evaluation and partnering. Negotiation is always the best resolution method to prevent disputes from happening, as it is requiring less time and saves cost down the road. Risk allocation promotes balanced risk distribution among the contracting parties. Steen [47] stated that allocating fair contract risk is one of the main ways to prevent litigation and solve construction disputes. An unfair shifting of risk later causes the parties to spend more time and effort finding ways to stay afloat in business [47]. Early non-binding neutral evaluation could be an alternative. The neutral entity is chosen by the parties to resolve the dispute with no intention to be biased [16]. This requires preselecting an interdependent "neutral" entity to serve the parties as an observer, fact-finder and dispute-resolver for as long as the construction is in process [47]. Some clients choose partnering to be their resolution method. Steen [47] recommended building teams as a key to prevent disputes. Building teams helps improve cooperation and coordination among different parties and helps to establish a better understanding between the parties [47].

### 4.2. Early Resolution Methods

Early resolution methods attempt to reach a satisfactory and acceptable solution to both parties, in which they seek to minimize the disputed amount (usually $ volume) or prevent moving to a more expensive and time-consuming method [15]. Once a dispute occurs, companies have various choices in picking an early resolution method according to their preference. Usually, disputants settle their

issues in the early stages by conciliation. This arises out of a clause in a construction contract whereby the parties agree to attempt to resolve their disputes through pacification and appeasement. The clause requires a conciliator, appointed by an agreement between the parties or by a specific institution. In addition, mini-trail or executive tribunal could be an alternative [9]. This process involves a panel, which consists of senior management representatives from each party as well as a neutral third party, or mediator who has been selected by the members of the organizations involved in the dispute.

## 4.3. Late Resolution Methods

Late resolution methods are used in the last stages of the dispute occurrence. These approaches are used in case of the failure of both avoidance and early resolution methods. While these methods can be very effective and efficient in settling the disputes, they are very expensive and consume considerable time and effort. In large and international companies, usually, the disputants use dispute review boards (DRB) instead of arbitration to provide an efficient and cost-effective means of dispute resolution [23]. On the other hand, local and small companies usually refer to mediation as an alternative solution. This method is a voluntary non-binding process in which a mediator assists the parties to retain full control over resolving the dispute [23]. Adjudication is another voluntary nonbinding process in which a mediator assists the parties in achieving a negotiated settlement [23]. Arbitration and adjudication can take place in all companies as the contract can identify everyone's rights without referring to the court. However, if all these methods are not applicable, the parties have no choice but litigation in the court of law. Usually, disputants try to avoid litigation, as this stage of dispute settlement is very costly and time-consuming to all parties involved.

## 5. Results

### 5.1. Assessment of the Sources of Construction Disputes in the UAE

Table 4 shows the sources of construction disputes, ranked in terms of their weighted average score based on the survey results.

**Table 4.** Main construction disputes in the United Arab Emirates (UAE).

| Sources of Disputes | Weight | Rank |
| --- | --- | --- |
| Variations initiated by the owner (additive/deductive) | 4.06 | 1 |
| Obtaining permit/approval from the municipality/different government authority | 3.87 | 2 |
| Material change and approval during the construction phase | 3.83 | 3 |
| Slowness of the owner's decision-making process | 3.81 | 4 |
| Time limitation in the design phase | 3.72 | 5 |
| Lack of communication and coordination between parties during construction | 3.7 | 6 |
| Poor financing by the owner | 3.69 | 7 |
| Inadequate early planning of the project | 3.67 | 8 |
| Poor preparation and approval of drawings | 3.65 | 9 |
| Underestimation and incompetence of contractors | 3.63 | 10 |
| Low financing by the contractor during construction | 3.59 | 11 |
| Unsuitable leadership style of construction/project manager | 3.54 | 12 |
| Failure to make interim awards on extensions of time and compensating by the owner | 3.52 | 13 |
| Shortage and unproductive manpower | 3.48 | 14 |
| Modifying legislation and regulations | 3.46 | 15 |
| Poor supervision and site management | 3.44 | 16 |
| Inadequate or incomplete technical plans/specification | 3.43 | 17 |
| Poorly written contracts | 3.35 | 18 |
| Contradictory and inaccurate information in the contract documents | 3.26 | 19 |
| Poor design | 3.17 | 20 |
| Contract amendments | 3.17 | 21 |
| Differing site conditions | 3.13 | 22 |
| Inadequate site investigation | 3.11 | 23 |
| Poorly defined scope of work | 3.11 | 24 |
| Inappropriate weather conditions | 2.8 | 25 |
| Impact on locality in terms of noise, traffic, and pollution/contamination | 2.76 | 26 |
| Impact of local cultures and social values in the settlement of conflicts | 2.72 | 27 |

The highest ranked dispute is "variations initiated by the owner", with a weighted average of 4.06. Most owners have limited or very little knowledge about planning, engineering and management of construction projects. This lack of knowledge frequently leads to multiple variations throughout the construction process. Another reason could be the typical rush through design and construction to complete the project. The second ranked dispute is "obtaining permit/approval from the municipality/different government authority", with a weighted average of 3.87. Obtaining approvals consume a lot of time and leads to delays in project completion. Due to the construction volume in the UAE, obtaining approval may take longer than expected. These delays usually result in disputes. The third ranked dispute is "material change and approval during the construction phase", with a weighted average of 3.83. The owners and the consultants typically initiate changes in material specifications during construction, causing delays in additional negotiation and approval. These kinds of delays give rise to disputes.

The fourth highest ranked dispute is "slowness of the owner's decision-making process", with a weighted average of 3.81. Such actions halt the contractor's production process and result in disputes. The fifth highest ranked dispute is "time limitation in the design phase", with a weighted average of 3.72. If the designer is limited by time, the quality of design will be adversely affected. This lack of quality contributes to disputes. The weighted average of the top five disputes range from 4.06 to 3.72, which is not very wide, indicating a consensus among the respondents. In addition, the range indicates how these causes of disputes occur in the UAE with a high frequency.

Figure 1 shows the weighted average of each of the five dispute categories. The numbers represent the average of all causes of disputes within the category. The highest is the 'owner', followed by the designer, contractor, contractual and other categories.

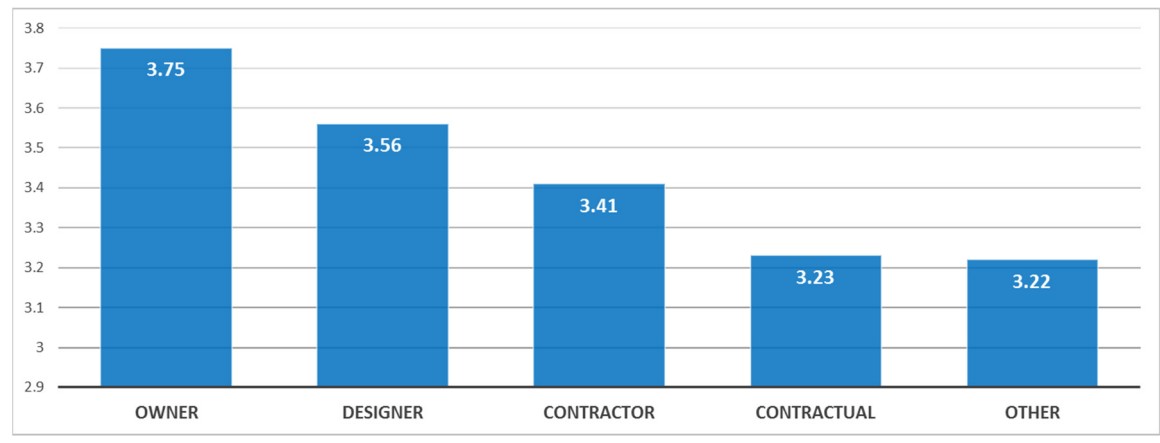

**Figure 1.** Weighted average of dispute categories in the UAE construction industry.

*5.2. Assessment of the Dispute Avoidance and Resolution Methods*

Based on the survey results, the effectiveness of each dispute avoidance and resolution method is determined as a weighted average of the responses. The results are summarized in Table 5.

Negotiation is found to be the most effective method in dispute avoidance, followed by risk allocation, early non-binding neutral evaluation and, finally, partnering. Negotiation is an effective avoidance method because of its ease of application. Construction organizations usually meet before the occurrence of potential disputes or issues to negotiate possible actions that can be taken before they happen. In addition, the negotiation method did not receive 'not applicable' response in the survey, meaning that it should be considered as an applicable method according to the respondents. Risk Allocation, Early Non-Binding Neutral Evaluation and Partnering methods received 'not applicable' responses by from some respondents. Partnering is the least effective method, due to the fact that not all parties are prepared to collaborate to resolve a dispute that has not occurred.

**Table 5.** Assessment of dispute avoidance and resolution methods.

| Method | Weighted Average |
| --- | --- |
| Dispute Avoidance Methods | |
| Negotiation | 4.15 |
| Risk Allocation | 3.5 |
| Early Non-Binding Neutral Evaluation | 3.3 |
| Partnering | 3.09 |
| Early Resolution Methods | |
| Negotiation | 4 |
| Conciliation | 3.59 |
| Mini-Trial/Executive Tribunal | 2.94 |
| Late Resolution Methods | |
| Negotiation | 3.72 |
| Arbitration | 3.31 |
| Mediation | 3.28 |
| Litigation | 3.28 |
| Adjudication | 2.87 |
| Dispute Review Board | 2.70 |

Negotiation was also found to be the most effective method as an early resolution method due to its ease and flexibility in reaching a resolution to the dispute at hand. (Normally, these kinds of negotiations are formal. Informal negotiations usually precede formal ones, to facilitate them.) Conciliation came in second place. Consulting with a neutral third party is favored because a neutral opinion on the matter of dispute is valued by the disputants. Finally, the 'Mini-trial' came in last because of reluctance to involve the court system and lawyers to solve the disputes. The involvement of lawyers and the court is always the last resort and the least preferred option to solve a dispute. However, if necessary, it is used.

As for the late resolution methods, the following methods are ranked from the most effective to the least effective, respectively, as Negotiation, Arbitration, Litigation Mediation, Adjudication, and Dispute Review Boards. Negotiation was shown to be an extremely effective method to avoid and resolve disputes, due to its ease of use, lack of complications, lack of involvement of other parties, and quickness. However, sometimes it is inapplicable depending on the severity of the issues causing the disputes. The second most effective method found is Arbitration, a method to solve disputes without the involvement of the court by consulting one or more arbitrators to determine what to do with the dispute at hand. This method helps to reach a solution faster than involving the court. The third most effective method is Litigation, in which is the court is involved to resolve the disputes according to the laws and regulations of the country. This method can be costly and time-consuming.

*5.3. Comparative Analysis*

The data were analyzed based on the different respondents' perspectives. The Spearman Rank Correlation Coefficient (RHO) was used to compare the resulting rankings. Table 6 shows the comparative results of the different categories. All the results are significant at the 0.01 level (two-tailed), with the exception of the contractor vs. consultant (in terms of resolution methods), which was significant at the 0.05 level (2-tailed). The Spearman Rank Correlation Coefficient (RHO) values were all positive and show strong agreement on the rankings of the causes of disputes and the dispute resolution methods.

**Table 6.** Comparative analysis.

| Categories | Spearman Rank | | |
| --- | --- | --- | --- |
| | RHO | Pval | Significance |
| **Causes of Disputes** | | | |
| Contractors vs. Consultants | 0.723 | 0.000 | Significant at the 0.01 level (2-tailed) |
| Years of Experience (0–10 vs. >10 years) | 0.757 | 0.000 | Significant at the 0.01 level (2-tailed) |
| Size (0–200M vs. >200M AED) | 0.72 | 0.000 | Significant at the 0.01 level (2-tailed) |
| Local vs. International | 0.754 | 0.000 | Significant at the 0.01 level (2-tailed) |
| **Resolution Methods** | | | |
| Contractors vs. Consultants | 0.681 | 0.011 | Significant at the 0.05 level (2-tailed) |
| Years of Experience (0–10 vs. >10 years) | 0.871 | 0.000 | Significant at the 0.01 level (2-tailed) |
| Size (0–200M vs. >200M AED) | 0.83 | 0.000 | Significant at the 0.01 level (2-tailed) |
| Local vs. International | 0.691 | 0.009 | Significant at the 0.01 level (2-tailed) |

Table 7 summarizes the results from the contractor and consultant perspectives.

**Table 7.** Sources of disputes—comparative results (contractors vs. consultants).

| Sources of Disputes | Contractor | | Consultant | |
| --- | --- | --- | --- | --- |
| | Average | Rank | Average | Rank |
| Time limitation in the design phase | 3.72 | 6 | 3.72 | 8 |
| Poor design | 3.21 | 20 | 3.12 | 23 |
| Inadequate or incomplete technical plans/specification | 3.34 | 17 | 3.52 | 13 |
| Poor preparation and approval of drawings | 3.83 | 4 | 3.44 | 14 |
| Material change and approval during the construction phase | 3.9 | 1 | 3.76 | 6 |
| Slowness of the owner's decision-making process | 3.79 | 5 | 3.84 | 2 |
| Inadequate early planning of the project | 3.66 | 8 | 3.68 | 10 |
| Failure to make interim awards on extensions of time and compensating by the owner | 3.62 | 10 | 3.4 | 15 |
| Variations initiated by the owner (additive/deductive) | 3.9 | 2 | 4.24 | 1 |
| Poor financing by the owner | 3.55 | 13 | 3.84 | 3 |
| low financing by the contractor during construction | 3.48 | 14 | 3.72 | 9 |
| Shortage and unproductive workers | 3.59 | 12 | 3.36 | 16 |
| Inadequate site investigation | 3.07 | 23 | 3.16 | 21 |
| Poorly defined scope of work | 2.97 | 24 | 3.28 | 17 |
| Poor supervision and site management | 3.31 | 18 | 3.6 | 11 |
| Unsuitable leadership style of construction/project manager | 3.48 | 15 | 3.6 | 12 |
| Underestimation and incompetence of contractors | 3.48 | 16 | 3.8 | 5 |
| Poorly written contracts | 3.62 | 11 | 3.04 | 24 |
| Differing site conditions | 3.1 | 22 | 3.16 | 22 |
| Contract amendments | 3.14 | 21 | 3.2 | 19 |
| Contradictory and inaccurate information in the contract documents | 3.28 | 19 | 3.24 | 18 |
| Obtaining permit/approval from the municipality/different government authority | 3.9 | 3 | 3.84 | 4 |
| Modifying legislation and regulations | 3.69 | 7 | 3.2 | 20 |
| Inappropriate weather conditions | 2.97 | 25 | 2.6 | 27 |
| Impact on locality in terms of noise, traffic, and pollution/contamination | 2.83 | 26 | 2.68 | 25 |
| Lack of communication and coordination between parties during construction | 3.66 | 9 | 3.76 | 7 |
| Impact of local cultures and social values in the settlement of conflicts | 2.76 | 27 | 2.67 | 26 |

## 6. Discussion

Construction is a complex process. There are several factors contributing to this complexity. One of the major factors is the interaction between different entities, such as the owner (client), the designers (architect/engineer) and the constructors (contractor/subcontractors) with conflicting objectives. Identification of the causes of disputes and mitigation (avoidance and resolutions) methods in the context of UAE is the subject of this paper. It is not too difficult to appreciate the fact that disputes in construction, where money is involved in great amounts, is inevitable. It is also not difficult to come up with a list of the main causes of disputes. In this paper, however, the causes of disputes and methods of mitigation were identified and ranked in order of their frequency, as perceived by the construction professionals in UAE. Professionals were drawn from the main entities, owners, designers

and constructors, and a survey was conducted. The results of the survey, as presented and discussed here, are the main contributions of this paper and are expected to enrich the body of knowledge on this subject.

As for the owners-related disputes, the highest weighted average is 4.06 for "variations initiated by the owner". As for designer-related disputes, the highest weighted average was 3.83 for "Material change and approval during the construction phase". The change in material occurs frequently, which can be initiated by any of the three main parties. Contractors can sometimes change materials in order to reduce cost; the owner, on the other hand, can do the same for decorative or aesthetic reasons or cost reduction. Sometimes, the specified material may not be available, or the market price of a specific material may become inhibitive and a replacement may become necessary. As for contractor-related disputes, the highest weighted average is 3.63 for "underestimation and incompetence of contractors". This cause of dispute is a consequence of contractors underestimating the scope and requirement of the job. It may even be due to the inability and lack of competency of the contractor to do the job. On the other hand, the lowest weighted average obtained was 3.11 for both "inadequate site investigation" and "poorly defined scope of work".

As for contractual-related disputes, the highest ranked source of dispute is "poorly written contract" by a weighted average of 3.35 and the "differing site condition" is ranked lowest, with 3.13. A contract is written between the parties, and details the rules and conditions for the construction project. A poorly written contract causes disputes between the parties, as the project is based on a weak agreement, leading to miscommunication between the parties. As for 'other' disputes, the highest rank is 3.87, for "obtaining permit/approval from municipality". The government regulations in obtaining approval vary in complexity and time requirement depending on jurisdictions. Often, the government regulations follow bureaucratic procedures for authorization of the approval required to proceed, thus causing unpredictable delays.

According to the survey results, the negotiation method is most recommended, whether it is during the avoidance, the early resolution or the late resolution phases. This is consistent with other studies [7]. A total of 90 to 95% of the construction claims are solved by the method of negotiation in the construction industry [7]. Negotiation is the number one process that is usually used when contractors try to deal with construction claims. If negotiation were not applicable, then it would be preferable to go with the risk allocation method during the avoidance phase, and the Conciliation method would be recommended during the early resolution phase. Regarding the late resolution phase, either the arbitration or mediation method, as these two methods both obtained the second highest responses.

The findings of this paper are expected to increase awareness in the UAE construction industry about the roles each major entity, the owners, the designers and the contractors, play in a construction project in relation to disputes and the methods of their mitigation. Their improved understanding of the causes of disputes is expected to have a positive impact by reducing the effects of disputes in construction.

## 7. Conclusions

At present, the UAE is one of the most vibrant countries as far as the construction sector is concerned. As a result, it is confronted with multiple issues including a significant number of construction disputes. This paper identifies the main causes of disputes in the UAE in terms of their frequency of occurrence. In addition, the paper identifies the main dispute avoidance and resolution methods and presents a comparative analysis of their effectiveness. The disputes are categorized in five different groups based on their sources. These are design, owner, contractor, contractual, and other. Owner-related causes are found to be the most predominant, as changes and modifications in the scope of the project and time taken by them to make decisions usually become grounds for dispute.

Mitigation techniques are investigated in three different phases of a construction project based on their appropriateness and effectiveness. They are avoidance, early resolution and late resolution.

Negotiation is found to be the most effective at all three stages. As far as the resolution methods are considered, conciliation, arbitration, mini-trial, and the use of Dispute Review Boards are found to be applied at various degrees. Litigation or settlement in a court is found to be the least desired by all entities.

**Author Contributions:** Conceptualization, S.E.-S. and I.A.; methodology, S.E.-S. and I.A.; validation, M.A., R.H., S.M. and O.E.-A.; formal analysis, M.A., R.H., S.M. and O.E.-A.; investigation, M.A., R.H., S.M. and O.E.-A.; resources, M.A., R.H., S.M. and O.E.-A.; data curation, M.A., R.H., S.M. and O.E.-A.; writing—original draft preparation, M.A., R.H., S.M. and O.E.-A.; writing—review and editing, S.E.-S. and I.A.; visualization, S.E.-S.; supervision, S.E.-S. and I.A.; project administration, S.E.-S. and I.A.; funding acquisition, S.E.-S. and I.A. All authors have read and agreed to the published version of the manuscript.

**Funding:** The work in this paper was supported, in part, by the Open Access Program from the American University of Sharjah. The APC was funded by grant number [OAP-CEN-091].

**Conflicts of Interest:** The authors declare no conflict of interest. The funders had no role in the design of the study; in the collection, analyses, or interpretation of data; in the writing of the manuscript, or in the decision to publish the results. This paper represents the opinions of the authors and does not mean to represent the position or opinions of the American University of Sharjah.

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
