# Peer review of "Construction Disputes in the UAE: Causes and Resolution Methods"

_buildings, doi:10.3390/buildings10100171_

Round 1
Reviewer 1 Report
This is a well-written paper, both in structure and in presentation. The reviewer has two recommendations for the authors. First, please report the population and the sampling approach of the survey in the manuscript. Second, the authors may conduct some statistical tests to analyze the data. For example, the authors may conduct Kruskal-wallis test to check whether there are significant differences in the ratings of respondents in different backgrounds. Analyses like this will make the paper stronger.
Author Response
The authors thank Reviewer 1 for the thorough and thoughtful review of the manuscript. The review helped the authors improve the paper significantly. Please find attached file for our point by point responses.

Reviewer 2 Report
Dear authors
My general comment is I like the paper - it is easy to read and a straight forward research task. Then I still have some critical comments to parts of the paper. I think it needs some more work before publishing. Here are my suggestions:
Abstract is OK
Introduction may be improved - in particular
- The background is OK and arguments for the topic are strong. It includes good reasons for presenting the context in which the research is done. Unfortunately, I do not think we get enough background from EAE construction and society. E.g. judicial system, building regulation and permit/approval system, even economic key facts may be relevant. Similarly the type of delivery model/procurement system and contract type.
- Mention that there is a list of resolution methods later in the paper.
- How big is the increase in conflict level? How do you measure that? In terms of delays and cost increase - may there be other reasons?
- page 2 line 68 - what is DRB?, line 83 ADR? (explain all abbreviations)
- page 2 line 87 - The main objectives are not clear to me. It seems you aim to identify root causes - but the text indicates you measure frequency. How does that line up? Second you want to identify appropriate methods - but you limit appropriate to mean effective (not appropriate for other reasons - tradition, legal, cultural etc.). Clarify.
Methodology
- p 3 line 99 includes the same problem as above - linking root cause/source with the argument of frequency (frequency indicates it happens often - but does not say why).
- Table 1 has one line repeated. The rest is OK. A basis of 54 respondents is not very strong. You may comment on that.
- How is the data analysis done?
Sources of construction disputes
- Table 2 is nice, but I expected to see "safety" in the contractor category and "contract type" in the contractual category. Why did they not occur?
- In the following text you should be consistent with the aim of the paper. Example p 5 line 129: "One of the most common disputes is ..." should say "...most common causes of disputes.." Right?
- Line 164: Who is responsible for obtaining permits? It is an external category - mention why this classification.
- Line 172 (p 6): "...lack of communication between parties during construction ..." - how is this an external dispute?
Dispute avoidance and resolution methods
- Table 3: Does Partnering include Alliancing?
- The overview of resolution methods given in Table 3 and 5 is useful. I would suggest mentioning this list in the introduction.
- Line 184: "Risk allocation promotes fair risk distribution..." I does so only if it is actually fair. I would suggest "Balanced risk allocation ..."
Results
- Table 4 is good, but make sure you are clear about how the ranking is done.
- Line 246 says "these causes are commonly occurring" - but did you actually ask the respondents about frequency of occurrence? I still cannot see the clear line between objective and the questions/results as mentioned first in the introduction.
- How did you extract the numbers in Figure 1 from the data?
- Table 5: How did you calculate the weighted average? Is it based on the score? I believe one resolution method is missing in the table: DRB.
- Line 261: There is always risk allocation (explicitly or implicitly). This should concern the quality of that allocation (balance, clarity etc.)
- Line 265 (p 10): Does this mean a strictly formal, or an informal style of negotiation?
- DRB is missing in the text - why is it less effective?
Discussion
- My general comment is: I cannot find any discussion. It is description of results and summary of observations concerning the data. There are no argument pro and con, no contrasts between theory and empirical results. The text or the heading may need rework.
- line 297 - does the contracts category include use of contract standards?
- line 301: "from one country to another" - yes but this is in UAE - not across.
- line 303: Why is it unpredictable (unanticipated)
- Line 306: you refer to "other studies" - what other studies?
- All in all the text on page 10 does not seem consistent with table 5 - check it.
Conclusion
- Again, you need to make sure the formulated objective, questions and results are consistent throughout so you formulate the conclusion towards the right objective (roots or frequencies?).
- Line 316: high level of occurrences .... is it high compared to what? More or less high compared to some base level of activity and conflict? (historically in previous periods)
References
- Dominated by old references - is there nothing new out there? You may comment on this in the theory section.
- If there are newer material - please include some key references.
Good luck
Author Response
The authors thank Reviewer 2 for the thorough and thoughtful review of the manuscript. The review helped the authors improve the paper significantly. Please find attached file for our point by point responses.
